# Introduction of Supervisor-Type Pediatric Hospitalists in a Tertiary Children’s Hospital: Experiences in a Hematology/Oncology Ward

**DOI:** 10.3390/children10081400

**Published:** 2023-08-16

**Authors:** Hong Yul An, Yun Jung Choi, So Hye Lee, Min Sun Kim, Hyun Jin Park, Bo Kyung Kim, Jung Yoon Choi, Hyoung Jin Kang, Saram Lee, Kyung Taek Hong

**Affiliations:** 1Department of Pediatrics, Seoul National University College of Medicine, Seoul National University Children’s Hospital, Seoul 03080, Republic of Korea; anhongyul@gmail.com (H.Y.A.); flubber224@gmail.com (Y.J.C.); mskim81@snu.ac.kr (M.S.K.); fionajin05@snu.ac.kr (H.J.P.); imkbk@naver.com (B.K.K.); choijy@snu.ac.kr (J.Y.C.); kanghj@snu.ac.kr (H.J.K.); 2Seoul National University Cancer Research Institute, Seoul 03080, Republic of Korea; 3Department of Hospital Medicine, Seoul National University Hospital, Seoul 03080, Republic of Korea; 4College of Nursing, Seoul National University, Seoul 08826, Republic of Korea; leesohye11@snu.ac.kr; 5Wide River Institute of Immunology, Hongcheon 25159, Republic of Korea; 6Department of Transdisciplinary Medicine, Seoul National University Hospital, Seoul 03080, Republic of Korea; 7Department of Medicine, Seoul National University College of Medicine, Seoul 03080, Republic of Korea

**Keywords:** hospitalist, pediatrics, personal satisfaction, child hospitalized, quality of health care

## Abstract

(1) Background: Hospitalists are healthcare providers who focus on hospitalized patients, but research on the roles of pediatric hospitalists is lacking. This study investigates the role of a supervisor-type hospitalist in a pediatric hematology/oncology ward at a tertiary children’s hospital, assessing the impact on satisfaction levels among patient caregivers, resident physicians, and nurses. (2) Methods: A retrospective analysis and online surveys were conducted to assess satisfaction levels before and after the introduction of hospitalists in the Department of Pediatric Hematology/Oncology at Seoul National University Children’s Hospital in the Republic of Korea. (3) Results: The introduction of hospitalists led to a 19.3% reduction in prescription error interventions over six months. Unexpected transfers to the intensive care unit decreased from 1.4% to 0.7% (*p* = 0.229). Patient caregivers reported elevated satisfaction levels with physicians (rated 8.47/10), and there was a significant enhancement in overall satisfaction among nurses (increasing from 3.23 to 4.23/5, *p* < 0.001). The majority of resident physicians (83.3%) expressed contentment with the hospitalist system, with 77% indicating an interest in transitioning to a hospitalist role. However, these resident physicians also expressed concerns regarding job stability. (4) Conclusions: Supervisor-type pediatric hospitalists have the potential to elevate satisfaction levels not only among patient caregivers but also among nurses and resident physicians, showing promise in improving medical care quality. Nonetheless, ensuring favorable perception and securing job stability within the hospitalist system are pivotal for achieving successful implementation.

## 1. Introduction

“Hospitalists” is a concept that has been a subject of active discussion within the US medical system since 1996. This system was established to alleviate the responsibilities of doctors with numerous outpatient patients, allowing for the delegation of inpatient care to professionals capable of efficiently managing patients. This approach enhances the value of skills and experience possessed by doctors dedicated to patient care [1].

In the United States, where the hospitalist system is most actively implemented, numerous studies have examined the role and status of hospitalists not only in hematology/oncology wards and in pediatric patients with complex severe disease, but also in sedation programs and COVID-19 treatment sites [2,3,4,5,6]. The hospitalist system has demonstrated a positive impact, especially in the field of hematology and oncology, where a substantial portion of patients require inpatient care. The system shows promise in playing a crucial role in cancer treatment [3]. To evaluate the impact of the hospitalist model on changes in the resident physicians’ working hours, a survey was conducted gathering data from both residents and existing hematopoietic stem cell transplant specialists at The Children’s Hospital of Philadelphia [2]. Consequently, the hospitalist system received favorable assessments across categories including continuity of care, experience level, and the establishment of a comfortable environment. Moreover, it was reported that the hospitalist system could serve as a valuable guide for fellowship training. Furthermore, studies have indicated the potential benefits of the hospitalist system in addressing reduced resident working hours, supporting residents, delivering high-quality patient care, reducing future costs, shortening hospital stays, and ensuring patient safety and quality.

As the hospitalist system gradually expands in Korea, studies on hospitalists have been reported in various medical departments, including internal medicine and surgery [7,8,9]. In a retrospective analysis, Han et al. found that 24 h full-time hospitalist care could independently predict lower in-hospital mortality of patients with acute medical issues compared to those receiving hospitalist care only during the day. They have also reported that 24 h hospitalist care facilitates timely treatment transitions and decreases unnecessary admissions to the intensive care unit (ICU) [7]. Jung et al. investigated the impact of a surgical hospitalist on postoperative outcomes and surgical costs. Patients admitted for surgery were divided into a hospitalist management group (HG, n = 298) and a non-hospitalist management group (NHG, n = 189). The study revealed that surgical hospitalist care resulted in reduced total hospital stays, fewer surgical complications, and a lower readmission rate of patients who underwent surgery, consequently lowering overall hospital costs [8].

However, there still is an insufficient number of studies on hospitalists at children’s hospitals in Korea. Korea’s medical insurance system is based on national health insurance for all citizens, which results in prices being established and regulated by the nation. Due to relatively lower prices for essential departments directly associated with vital organs, the preference of young doctors for these medical departments has seen a recent decline. In the Republic of Korea, for example, there are only 69 subspecialists in pediatric hematology/oncology across the nation, and, among them, not all are actively practicing. Given an annual incidence of approximately 1000 pediatric cancer patients, this underscores a significant scarcity of medical resources. To address this challenge, a hospitalist system was implemented. Given the context of tertiary children’s hospitals in Korea, where the severity and demand for specialized treatment have risen due to an increasing number of patients with severe and complex diseases, and considering the ongoing decline in the pediatric resident applications, it is necessary to explore the role of an appropriate pediatric hospitalist tailored for a tertiary children’s hospital, rather than merely adhering to the work patterns of internal medicine or surgery hospitalists. This study aims to examine the beneficial outcomes of introducing supervisor-type pediatric hospitalists in the hematology/oncology ward of a Korean tertiary children’s hospital, including the perspectives of patients, caregivers, and medical staff, with the goal of improving medical services.

## 2. Materials and Methods

The hypothesis of our study is that introducing supervisor-type pediatric hospitalists to a hematology/oncology ward of a tertiary children’s hospital could lead to improvements not only in medical-related indicators, but also in the satisfaction levels of patients, caregivers, and medical staff. This study involved an analysis of medical-related indicators and the administration of surveys to patient caregivers and medical staff (resident physicians and nurses) both prior to and following the implementation of hospitalists in the Department of Pediatric Hematology/Oncology at the Seoul National University Children’s Hospital. The pediatric hematology and oncology ward consists of a total of 28 beds, including 6 sterile rooms, and offers medical services such as chemotherapy, hematopoietic stem cell transplantation, and complication management. In March 2021, two supervisor-type hospitalists were assigned to guide and educate resident physicians in the role of primary care, as well as to perform patient care and counseling. The study was conducted in two phases. Initially, we retrospectively analyzed indices such as the average length of hospital stay, occurrence of unexpected ICU transfers, and the number of cases of cardiopulmonary resuscitation (CPR). Subsequently, we administered surveys through an online platform to 136 caregivers of patients who had been hospitalized for more than 3 days in the pediatric hematology and oncology ward, along with 29 nurses and 61 resident physicians who had worked in the same ward. This study was a pilot investigation and was designed as a single-arm study. We aimed to include the maximum feasible number of patients and caregivers within the defined research period at our institution. If consent was obtained, we enrolled the maximum number of resident physicians and nurses. Caregivers with experience in hospitalist-run wards were given a medical satisfaction survey, while medical staff were surveyed to assess their overall satisfaction following their experience working with hospitalists. The explanation and consent process for the medical staff-related research were conducted by a researcher with no interest in the study’s subject. Anonymous responses were gathered through the survey platform. Quantitative data were analyzed using IBM SPSS statistics 22.0 (IBM: Armonk, NY, USA). Categorical data were analyzed using the chi-square test and Fisher’s exact test, while continuous data were assessed using independent t-tests and the Mann–Whitney test. All visualized data, including table and figures, complied with copyright laws and regulations. We have taken utmost care to ensure that all content included in the paper adheres to the necessary legal and ethical standards.

This study was approved by the Institutional Review Board (IRB) of Seoul National University Hospital on 8 February 2021 (IRB number: 2101-102-11898). Given that this study exclusively pertains to the distribution of results from an online survey and does not encompass any sensitive identification and medical records, the necessity for written informed consent was accordingly waived.

## 3. Results

### 3.1. Comparison of Medical Practices in the Ward before and after Hospitalist Implementation

To compare and evaluate the medical practices before and after the introduction of hospitalists, we conducted a retrospective analysis of the following parameters: total number of inpatients from March to August 2020 before hospitalist placement in the pediatric hematology and oncology ward, and from March to August 2021 after placement, the number of hospitalizations and route of admission, average length of stay in the ward, the types of diagnosis, and the frequency of interventions to correct prescription errors (Table 1). The term “interventions to correct prescription errors” refers to instances where drug administration order was reviewed by pharmacists, and checked for errors such as incorrect dosages.

No significant statistical differences were noted in terms of the number of patients and hospitalizations, admission routes, the duration of hospitalization, and types of diagnosis. Comparing the period prior to the assignment of the hospitalists, the frequency of interventions to correct prescription errors decreased by 42 cases (from 218 cases to 176 cases) over six months, averaging 0.23 cases per day post hospitalist assignment. The rate of unexpected ICU transfer also showed a decreasing trend from 1.4% to 0.7% (*p* = 0.229). There was no case of CPR performed in the ward for six months following the hospitalist assignment. While there were six and two deaths in the ward before and after the hospitalists, respectively, both sets of cases were attributed to disease progression in patients.

### 3.2. Patient and Caregiver Survey

Between July 2021 and October 2021, a survey was conducted involving 136 caregivers of patients hospitalized in a pediatric hematology and oncology ward. Among the 70 caregivers who participated, a response rate of 51.5% was achieved, revealing an average hospitalization duration of 25.2 days. Additionally, 50% of the patients had previously received treatment at other hospitals, and 15 of these patients (21.4%) had received hematopoietic stem cell transplantation. The caregivers who responded to the survey reported an average satisfaction level of 8.44 out of 10 for the physicians, with the satisfaction factors detailed as shown in Figure 1.

In addition, caregivers were surveyed regarding their comparative assessment of inpatient treatment experiences between the current hospitalist-run ward and the previous general ward. Among the 38 caregivers with a previous hospitalization history, the average satisfaction level improved to 8.47 points on the scale of 0 points at the worst satisfaction level to the full scale of 10 points. The areas showing improvement are detailed as illustrated in Figure 2.

Patient caregivers specifically highlighted the hospitalist ward system’s strengths in understanding the patient (57.9%), providing prompt and appropriate treatment (57.9%), and providing a comprehensive explanation and interview (55.3%). Items with relatively low response rates included the efficient management of admission and discharge (15.8%), proficiency in treatment and technique (13.2%), and organized discharge progress (10.5%). None of the caregivers responded that the system had worsened compared to before the implementation of the hospitalist system.

Despite being admitted to the hospitalist-run ward, 44.6% of the caregivers answered that they were not familiar with the hospitalist program. 72.3% of caregivers expressed support for adopting the hospitalist program, while only 15.4% responded negatively to the system. The most anticipated aspect of the system’s implementation was the promptness of appropriate action (55.4%), and the aspect of greatest concern was the communication issues with outpatient physicians (60.0%). Among caregivers, 90.8% stated their willingness to pay an additional charge beyond the current fee. As for the amount of additional charge that they could afford, caregivers answered 21.5% for less than 1000 KRW, 46.2% for 1000 KRW to 5000 KRW, 16.9% for 5000 KRW to 10,000 KRW, 6.2% for 10,000 KRW to 30,000 KRW, 0% for 30,000 KRW to 50,000 KRW, and 3.1% for more than 50,000 KRW on a daily basis (1 USD = 1315 KRW as of 01 March 2023).

### 3.3. Nurse Survey

A total of 29 nurses working in the pediatric hematology and oncology ward were surveyed. Of all 29 nurses, 16 (55.2%) had worked for more than 5 years, and 62.1% of the all respondents replied that their current workload was burdensome. Of the 29 nurses who were subject to investigation, 13 (44.8%) responded to the indicated survey. In all items comparing resident physicians and hospitalists, the hospitalists were regarded as more competent in the overall evaluation of performance. When evaluated on a five-point scale, the following responses showed a positive perspective towards the hospitalist program: smooth communication (3.77 vs. 4.15, *p* = 0.24), prompt response to nurses’ needs (3.31 vs. 4.15, *p* = 0.005), help with patient care (3.62 vs. 4.46, *p* = 0.014), prescription without error (2.54 vs. 3.46, *p* < 0.001), improvement in nursing environments (3.69 vs. 4.00, *p* = 0.104), appropriate medical treatment (3.31 vs. 4.38, *p* < 0.001), proficiency in interventional procedures (2.69 vs. 3.85, *p* < 0.001), and overall satisfaction (3.23 vs. 4.23, *p* < 0.001). As for the negative question, hospitalists were responded to with fewer points on conflicts with medical staff (2.62 vs. 1.92, *p* = 0.006), and 10 out of the 13 respondents (76.9%) thought that the system was helpful in resolving conflicts.

In the evaluation of nurses’ overall satisfaction with the hospitalist system, 10 of the 13 respondents (76.9%) answered that it was helpful. Excluding the reduction in nurses’ working hours, affirmative responses were given for items regarding increased patient satisfaction, reduced anxiety in patient care, and enhanced efficiency in nursing work (Figure 3). Regarding specific aspects of inpatient care, 69.2% of nurses indicated that the system helped to improve inpatient management and care, and others responded affirmatively that the system helped to increase satisfaction with patients and caregivers, and to share professional knowledge. None of the nurses responded that the program was unhelpful. When asked about the possibility of future collaboration, 11 out of 13 nurses (84.6%) responded positively. Nurses emphasized that the paramount role of hospitalization specialists lies in providing dedicated patient care. As for the prerequisites to vitalize the hospitalist program, the following responses were provided: fortifying the authority and responsibility for medical treatment (76.9%), enhancing job security (e.g., guaranteed social position) (53.8%), and improving working conditions (e.g., wages) (30.8%).

### 3.4. Resident Physician Survey

Of the 61 pediatric residents with prior exposure to the hospitalist-run ward, 30 participated in the survey, yielding a response rate of 49.2%. The survey was conducted for all residents, including nine from the first year, four from the second year, six from the third year, nine from the fourth year, and two who did not respond. When evaluating changes in the working environment after the implementation of the hospitalist program, 60.0% to 86.7% of responses indicated a positive change across all aspects. Negative responses, such as ‘no difference’, were minimal, accounting for only 10%. This suggests a favorable impact of the hospitalist program on the working environment, especially noticeable among first-year resident physicians. Regarding the role of hospitalists in inpatient care, the highest response rates were observed for ‘improvement of inpatient management and quality of care’ (76.7%) and ‘sharing expertise’ (76.7%) (Figure 4).

Most items related to job satisfaction evaluation, such as ‘smooth performance of management, such as prescription and procedure’ (4.40 on average) and ‘quick response to necessary matters’ (4.23), showed a very high level of satisfaction with scores of 4.0 or higher. Across all items, the positive response rates exceeded 80%, and, notably, no negative responses were recorded, indicating a substantial job satisfaction level within the hospitalist program. Only 1 out of 30 (3.3%) respondents answered ‘Yes’ when asked whether there was a conflict with the hospitalist, and most respondents answered ‘No’ (93.3%), indicating a near absence of conflict. Regarding specific instances of conflict, respondents pointed out potential issues during night duty transition (26.7%) and potential vertical conflicts with hospitalists (13.3%). Notably, responses about conflict resolution were relatively high for ‘independence of the hospitalist-run ward’ (26.7%) and ‘recruitment of hospitalist personnel’ (26.7%). Regarding whether the hospitalist contributes to the education and training of residents, 70% of respondents answered ‘contributes’. ‘Lecture’ (43.3%) and ‘Patient’s physical examination’ (40.0%) were frequently answered for the education method, and practical training in the field was found to be relatively preferred. The reasons that the hospitalist system is considered helpful for the education of majors turned out to be ‘Parallel education with practical work in the field’ (76.7%), ‘one-on-one customized training’ (56.7%), and ‘education on various cases’ (50.0%), implying the core strength of the program as practical education in the field. Responses suggesting that the hospitalist program had a limited impact on resident education often cited ‘Insufficient training time’ (73.3%), underscoring the need to secure adequate training time for more effective resident education in the future. Overall satisfaction with the hospitalist system was notably high, with 83.3% of respondents expressing contentment, and 90.0% indicating a preference to work with hospitalists as residents in the future.

In response to the query regarding their willingness to pursue a career as a hospitalist following the completion of their residency training, 23 respondents (76.7%) answered ‘yes’ (Figure 5A). All nine fourth-year residents responded that they were considering hospitalist as their future career. Regarding the support necessary to further vitalize the hospitalist program, the most frequently cited responses were ‘strengthening job security’ (73.3%), ‘enhancing authority and responsibility for treatment’ (63.3%), and ‘improving labor condition (ex. wages)’ (50.0%). When asked about the factors motivating their choice of a hospitalist career path, respondents highlighted ‘the opportunity to continue treating inpatients as a professional’ (56.5%), ‘the role serving as a valuable intermediate step towards a future hospital-based position’ (56.5%), and ‘favorable working environment and compensation’ (52.2%). Conversely, all seven respondents who were unwilling to choose hospitalist as a career path answered ‘job insecurity’ as the biggest obstacle to choosing hospitalist as their future career (Figure 5B)

## 4. Discussion

This study encompassed not only the overall change in medical practices, but also an exploration of the satisfaction and system-related viewpoints of patients, caregivers, nurses, and resident physicians regarding the hospitalist program in the pediatric hematology and oncology ward at a tertiary children’s hospital in Korea. While it remains an undeniable fact that the introduction of the hospitalist program is a direction for better inpatient care in the future, the program’s management approaches can be diversified, considering the distinctive disease profiles and departmental dynamics across various medical settings. Given the scarcity of hospitals equipped to manage pediatric patients with severe diseases of hematology and oncology, coupled with the declining application rates for pediatrics residencies, a supervisor-type hospitalist program may be the resolution to provide high-quality inpatient care experience while providing residency education simultaneously. Notably, recent research underscores the significance of mutual feedback between hospitalists and trainees, particularly from residents [10].

While the overall positive perspectives regarding the hospitalist program have been substantiated, it is important to acknowledge the existence of prior negative perceptions [11,12]. Gunderman et al. had expressed concerns that the hospitalist program could threaten the existing medical care system by diminishing doctor–patient reliance [11], and there also is a study that predicted the high cost of cooperation between the hospitalist program and the existing medical care system [12]. However, a survey encompassing over 8000 patients revealed no significant disparity in patient satisfaction levels between the existing medical care model and the hospitalist program. This finding indicated that the concern regarding a lack of continuity in outpatient and inpatient treatments, as highlighted in previous studies, is not statistically significant [13].

In this study, comparisons indicated a reduction in the number of prescription error interventions and unexpected ICU transfers; however, these decreases did not reach statistical significance (*p* = 0.229). Unlike findings in studies conducted in other internal medicine and surgical departments, there was no observed decrease in the length of hospital stay. This is likely due to two factors: first, a shortage of medical resources such as hospital beds in our institution’s pediatric hematology and oncology department, which may lead to the inefficient use of available resources; and second, the inherent limitations of making indirect comparisons between different time periods within a single institution [7,8,14].

The study’s survey confirmed high levels of satisfaction with the hospitalist program among all subjects, including patient caregivers, nurses, and resident physicians. Notably, they expressed particular contentment with aspects of understanding patients and providing sufficient explanations and interviews. The role of hospitalist was found to have been performed appropriately, aligning with findings from previous studies [2,3]. Establishing rapport with patients and promptly building a sense of trust is an essential competency of hospitalized doctors [15]. On the other hand, the patients’ most dissatisfied items were related to the management of admission and discharge, as well as the proficiency in treatment and procedures. The insufficient management of inpatient and discharge processes could potentially stem from the hospitalist’s limitations in efficiently handling medical resources like hospital beds. Other medical staff such as nurses and residents also showed a high satisfaction with the program, with items such as ‘smooth communication’ and ‘prompt response and treatment’ scoring highly, and the overall satisfaction with the program was also highly scored. This is consistent with a previously published study that found a positive correlation between inter-disciplinary interventions and the hospitalist system, leading to improved teamwork and communication, as evaluated by nurses, subsequently increasing nurse satisfaction [16]. Furthermore, conflicts among medical staff showed an improved aspect compared to the existing single physician model, and there was also less conflict between the residents and the hospitalists, highlighting the program’s positive impact on treatment dynamics, as reported in the previous study [2,3].

In contrast to previous reports, this study employed a unique supervisor-type hospitalist model, differing from the direct treatment approach often associated with hospitalists. This novel approach, which has not been previously documented in Korea and is also rarely observed internationally, aligns with the distinctive attributes of our institution. Our hospital frequently receives referrals for patients with elevated disease severity or intricate conditions, primarily emphasizing the education of resident doctors [9,17]. The hospitalist system was previously reported as effective for patients with comorbidities requiring clinical monitoring, or for patients in need of a complex discharge plan; thus, such a model was chosen for the study. This model is also related to the works of residents, and it will also enable ‘complementing the reduced working hours of residents’, providing ‘resident support’, and ensuring ‘qualified patient care’. According to Hinami et al., although hospitalists have high job satisfaction, burnout symptoms are common among them. In order to increase the satisfaction of hospitalists and minimize their consumption, their work environment should be reorganized, and their personal times and compensations should be guaranteed. Physicians and hospitalists will be paired complementarily according to the supervisor-type model, and hospitalists will be able to continue working without being exhausted [18]. Furthermore, further research is required to address the staffing of hospitalists for night shifts and to enhance patient safety [19].

Despite these advantages, the contemporary domestic hospitalist program requires continuous improvement, as previously undertaken in the USA [20]. Furthermore, considering the declining rate of applications to pediatric residency due to low birth rates and low medical fees, it is imperative to enhance the pediatric hospitalist program in order to establish a positive cycle that ensures continued care for children within the domestic context. Currently, the hospitalist system is only operational in a few hospitals. This situation can be attributed to factors such as insufficient job stability, authority, and responsibility, and dissatisfaction with wages and work conditions, as indicated by nurses and residents in a survey. Although the reason for choosing hospitalist as a future career may vary, those residents uninterested in such a path have cited ‘job insecurity’ as a major hurdle, underscoring the significance of job stability concerns in their career decision-making process.

## 5. Conclusions

In conclusion, this study shares the operational experience of a supervisor-type, pediatric, hospitalist-run ward in the field of pediatric hematology/oncology at a tertiary children’s hospital in Korea. This study has provided a positive view of this system from the perspective of patients, caregivers, nursing staff, and resident physicians who have experienced improved working environments. Furthermore, this study suggests that caregivers’ willingness to receive more qualified care with additional charges could be a solution to realizing the hospitalist management fee at children’s hospitals. Additionally, policies should be implemented to strengthen job security through appropriate inpatient care performance evaluation, close communication with existing outpatient medical staff, and the strengthening of treatment independence to facilitate the mutual development of the existing medical staff and hospitalists. A continuous discussion regarding the appropriate hospitalist program for future children’s hospitals will remain crucial.

## Figures and Tables

**Figure 1 children-10-01400-f001:**
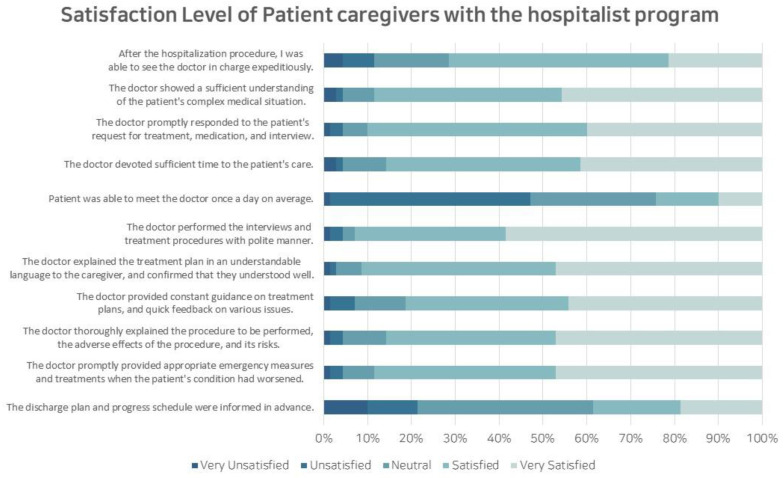
Satisfaction level of patient caregivers with the hospitalist program.

**Figure 2 children-10-01400-f002:**
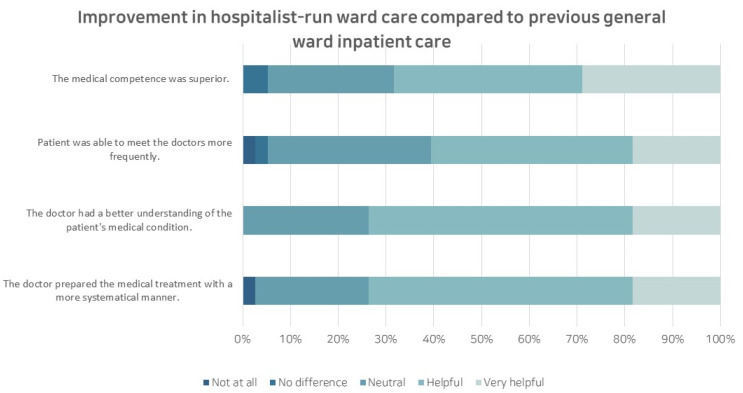
Improvement in the hospitalist-run ward care compared to the previous general ward inpatient care.

**Figure 3 children-10-01400-f003:**
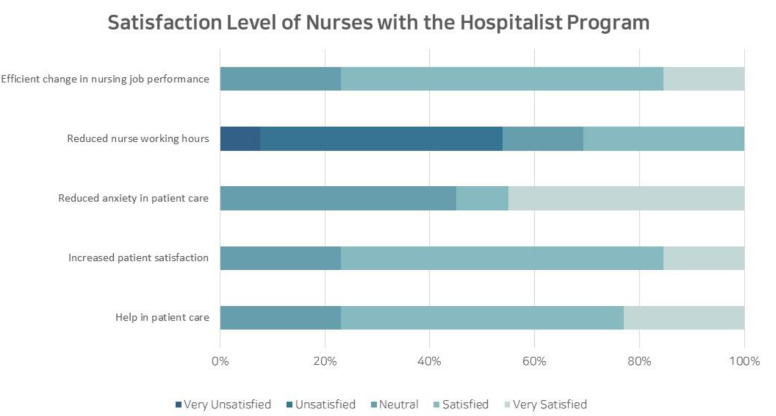
Satisfaction Level of Nurses with the Hospitalist Program.

**Figure 4 children-10-01400-f004:**
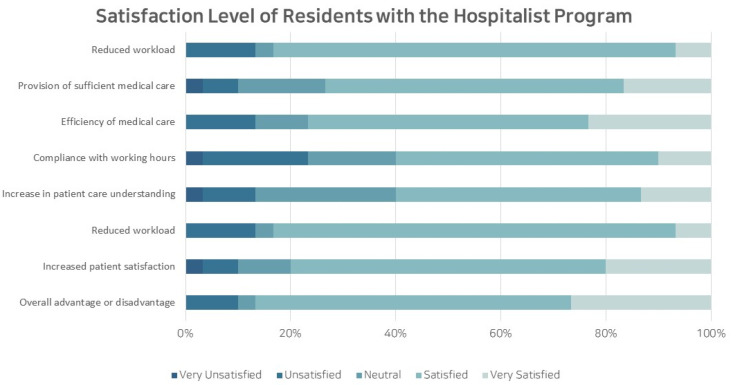
Satisfaction Level of Residents with the Hospitalist Program.

**Figure 5 children-10-01400-f005:**
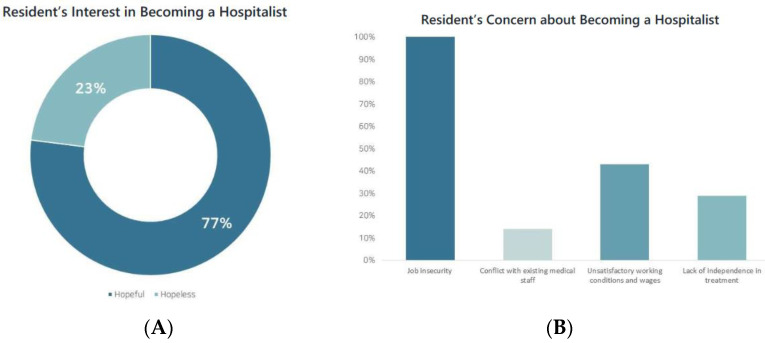
(**A**) Resident’s Interest in Becoming a Hospitalist. (**B**) Resident’s Concern about Becoming a Hospitalist.

**Table 1 children-10-01400-t001:** Comparison of Medical Practices in the Ward Before and After Hospitalist System.

		Period		
Characteristics		Before Hospitalists (March to August 2020)	After Hospitalists (March to August 2021)	*p*-Value
Total patients, n		399	409	
Total hospitalization, n		739	704	
Admission via emergency room, n (%)		254 (34.4)	257 (36.5)	0.397
Average length of stay per admission, day		10.4	10.1	0.665
Total hospitalization days, day		7704	7080	
Diagnosis, n (%)				0.109
	Acute lymphoblastic leukemia	96 (13.0)	73 (10.4)	
	Acute myeloid leukemia	49 (6.6)	48 (6.8)	
	Malignant lymphoma	42 (5.7)	55 (7.3)	
	Brain tumor	93 (12.6)	83 (11.8)	
	Other solid tumors	350 (47.4)	364 (51.7)	
	Others	109 (14.7)	81 (11.5)	
Intervention to correct prescription errors, n		218	176	
Unexpected ICU transfer, n (%)		10 (1.4)	5 (0.7)	0.229
CPR events, n		1	0	
Mortality, n (causes)		6 (all disease progression)	2 (all disease progression)	

CPR, cardiopulmonary resuscitation; ICU, intensive care unit; n, number.

## Data Availability

Data are available upon request to the corresponding author.

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
