# Peer review of "Introduction of Supervisor-Type Pediatric Hospitalists in a Tertiary Children’s Hospital: Experiences in a Hematology/Oncology Ward"

_children, 2023, doi:10.3390/children10081400_

Round 1

Reviewer 1 Report

According to the manuscript, there still is a deficient number of studies on hospitalists at children's hospitals in Korea, therefore, the study is important for the Korean health sector. However, the authors should provide more information about the essence and methodology of hospitalists to convey knowledge that is relevant to international practice. In my opinion, the applied study design is not suitable for investigating the research objectives and revealing the beneficial effects of introducing hospitalists. In my opinion, a randomized, controlled group design would have been more effective.

The title of Figure 1 is: "Satisfaction Level of Patient Caregivers with the Hospitalist Program," however, the table does not illustrate data related to satisfaction. It seems to me that the figures are misaligned or misplaced.

I believe that the study needs to be clarified, supplemented, and corrected in several aspects.

Reviewer 2 Report

page 1, line 38, replace "in deals with" with "concerning"

page 2, line 51, add reference!

Please add the main source of financing of the health system in Korea! I suppose contributions if the system is based on mandatory health insurance. Add in one paragraph the organization of health system according to the levels of health care.  What is the number of paediatricians per 1000 inhabitants? I suppose there are lacking.

method, add one sentence about the type of sampling design! How the sample size was determined?

page 4, Figure 1 should be table 2. Please, make the appropriate table as table 1 (the look of the tables should be the same).

Discussion, obtained results must be discussed with more appropriate references especially regarding satisfaction level domains of patient caregivers, residents and nurses! (only 15 refs in the whole manuscript)

Round 2

Reviewer 1 Report

The authors have addressed major concerns about previous version.

Author Response

We sincerely appreciate your insightful assessment of our paper.

Reviewer 2 Report

Minor comments:

Discussion

Page 9, lines 314-315. "The role of hospitalist was found to 314 have been performed appropriately, aligning with findings from previous studies". Please, add those studies, that is, references!

Page 9, lines 330-331. Put "....in the previous studies (2, 3)."
